# CoM-V2I: Communication-Efficient Multi-modal Cooperative Perception via Codebook Pruning and Multiscale Fusion

## Abstract

Cooperative perception, which fuses sensory information from multiple agents to enhance individual agent's perception ability, has emerged as a promising approach to overcome the limitations of single-agent line-of-sight sensing. However, a significant challenge lies in economically deploying sensors across agents while minimizing communication costs and maintaining strong perception performance. To address this challenge, we propose CoM-V2I, a novel framework for Communication-efficient Multimodal Vehicle-to-Infrastructure (V2I) cooperative perception. In CoM-V2I, the road infrastructure is equipped with a high-resolution LiDAR sensor, while vehicles are fitted with cost-effective multi-view cameras to balance performance with economic feasibility. We introduce a residual vector quantization-based codebook representation method to improve communication efficiency by compressing bird's eye view (BEV) feature maps into lightweight indices before transmission. We also propose a codebook pruning method that reduces codebook size by removing low-importance code vectors and combining high-similarity ones, thereby decreasing communication costs with minimal impact on perception performance. Furthermore, we propose a multiscale fusion mechanism that progressively integrates multimodal BEV feature maps from the infrastructure and vehicles, which have different spatial resolutions in a coarse-to-fine manner. Experimental results on the V2X-Real and V2X-Sim datasets demonstrate that the proposed CoM-V2I framework outperforms existing baselines in terms of perception accuracy and communication efficiency.

## 1 Introduction

The efficiency of transportation systems has improved significantly with recent advances in machine learning for autonomous driving (Yin et al., 2025). Perception is a key component of this advancement, which is the process of interpreting raw data from onboard sensors, e.g., LiDAR, cameras, and radar, to create a structured and machine-readable representation of the surrounding environment (Kiran et al., 2021). However, perception systems based on a single agent face several challenges, especially a limited perceptual range and vulnerability to occlusions (Li et al., 2024). To overcome this bottleneck, cooperative perception (CP) has emerged as a promising solution that enables an agent to broaden its perceptual area by fusing perceptual information from other connected agents via vehicle-to-everything (V2X) communication (Chen et al., 2019a). Early CP methods were primarily based on 3D point cloud data from LiDAR and can be broadly divided into three categories: early, late, and intermediate fusion (Arnold et al., 2022). Intermediate fusion became the mainstream approach in subsequent research, as it allows for the transmission of rich feature representations with low communication costs (Prakash et al., 2021). Despite its effectiveness, the high cost of LiDAR has been a significant limitation to its widespread adoption in vehicular CP systems. For this purpose, transformer-based methods like BEVFormer (Li et al., 2025) and cross view transformer (CVT) (Zhou & Krähenbühl, 2022) have been developed to generate bird's-eye-view (BEV) representations from multi-view cameras, achieving 60% to 70% of the performance of their LiDAR-based counterparts. Furthermore, multi-view camera-based CP has been shown to offer an acceptable trade-off between performance and cost (Xu et al., 2022a). To improve the scalability of CP beyond systems limited to homogeneous sensors, recent studies have begun to focus on multi-

modal CP (Xiang et al., 2023) and on scenarios involving various types of agents, such as vehicles and road infrastructures (Zhou et al., 2025), while several critical issues remain. A primary challenge is achieving a favorable trade-off between cost and performance in multi-modal CP among heterogeneous agents.

Communication efficiency is another critical issue in CP, as high transmission latency can significantly deteriorate performance by causing pose errors that disrupt the feature fusion process (Han et al., 2023). Many studies aim to reduce communication costs through feature filtering and selection techniques (Yang et al., 2023b; Liu et al., 2020; Yang et al., 2023a; Hu et al., 2022). Shannon's theorem suggests that acceptable communication latency can be maintained if the transmitted features are compressed sufficiently to fit within the reduced channel capacity caused by channel fading (Tang et al., 2025). Codebook-based approaches can effectively compress transmitted features by representing them with compact index matrices (Hu et al., 2024). However, a single codebook has limited representational capacity, thus more efficient representation methods with low-latency are urgently needed for CP.

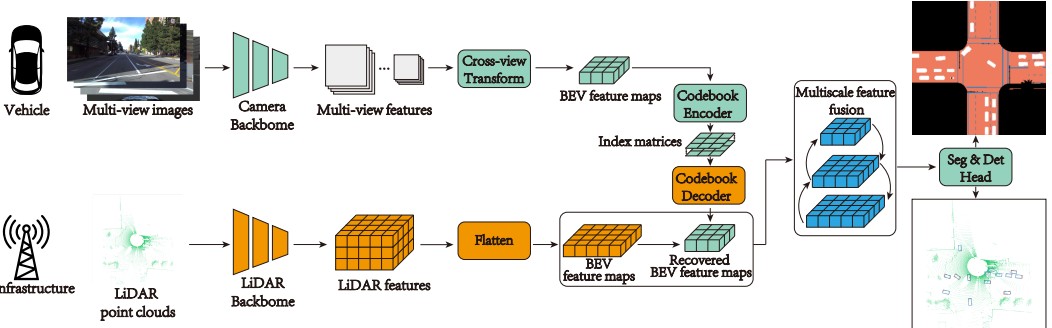

Figure 1: The overall framework of CoM-V2I.

To address the above challenges, we propose CoM-V2I, a novel multi-modal CP framework among road infrastructures and vehicles. The proposed framework is illustrated in Fig. 1. For economic consideration, we deploy a high-precision and expensive LiDAR only on the infrastructure, while equipping all vehicles with more affordable multi-view cameras. Point clouds and images are processed into BEV feature maps by their respective pipelines. A codebook encoder, composed of multiple codebooks, is used to represent the camera-based BEV feature maps as compact indices recursively via residual vector quantization (RVQ), incorporating a codebook pruning method that shortens indices via decreasing code vectors to reduce communication costs. The infrastructure then reconstructs the transmitted features using a shared decoder, implements a multiscale fusion method to all BEV feature maps and outputs predictions using task-oriented heads. The main contributions of this paper are as follows.

- We propose the CoM-V2I framework for multimodal CP cross vehicles and the infrastructure with efficient communication via residual codebook representation. A multiscale fusion strategy is developed and incorporated into the CoM-V2I for combining BEV feature maps of various modalities and resolutions.
- We devise an effective codebook pruning method that optimizes communication costs by reducing representation indices through removing low-importance code vectors and merging those with high similarity.
- We conduct extensive experiments and ablation studies on the V2X-Real and V2X-Sim datasets. The results demonstrate the state-of-the-art performance of our proposed CoM-V2I framework and the effectiveness of our codebook pruning method.

## 2 RELATED WORK

**Birds eye view representation.** BEV representation aims to reconstruct the surrounding environment into a machine-readable format from a top-down perspective using sensor data, e.g., point clouds and images(Reiher et al., 2020b). Early works were primarily LiDAR-based, projecting

3D point clouds onto a 2D BEV grid to accelerate processing, as seen in methods like PIXOR (Yang et al., 2018) and PointPillars (Lang et al., 2019), while the high cost of LiDAR has limited its widespread application. Consequently, recent research has focused on generating BEV representations from more economical multi-view cameras (Zhao et al., 2024). This has led to a variety of approaches, including geometric methods like Inverse Perspective Mapping (IPM) (Reiher et al., 2020a), depth-based techniques such as Lift-Splat-Shoot (LSS) (Philion & Fidler, 2020), and more recent Transformer-based architectures like BEVFormer (Li et al., 2025) and CVT (Zhou & Krähenbühl, 2022). To leverage the strengths of both sensors, multi-modal fusion methods have been developed to combine LiDAR and camera features using either convolutional neural networks (CNNs) (Liu et al., 2023b; Cai et al., 2023) or Transformers (Li et al., 2022b; Bai et al., 2022). However, the challenge of effectively fusing features with different spatial resolutions has been underexplored in prior work. In this paper, we propose a novel multiscale fusion method specifically designed to address this gap.

**Cooperative perception.** CP aims to improve the perception quality and range of an agent by fusing sensory information from multiple connected vehicles and road infrastructures (Liu et al., 2023a). Early works primarily focused on LiDAR-based CP, sharing information via early, intermediate, or late fusion strategies (Chen et al., 2019b; Wang et al., 2020; Rawashdeh & Wang, 2018). Among these, intermediate fusion became the mainstream approach due to its favorable trade-off between performance and communication costs. Driven by cost-efficiency and recent advances in camera-based methods, subsequent research has explored sharing BEV feature maps constructed from multi-view images (Xu et al., 2022a; Song et al., 2024). This has also broadened the scope of CP to include multi-modal data across both homogeneous and heterogeneous agents (Xiang et al., 2023; Zhou et al., 2025). A key challenge in CP is communication latency, which can degrade performance by causing issues like pose errors (Lei et al., 2022). To mitigate this, many recent works aim to reduce communication costs via message filtering and selection (Yang et al., 2023b;a; Hu et al., 2022). A more advanced approach, proposed in Codefilling, uses a shared codebook to represent features, transmitting only compact indices instead of the full feature maps (Hu et al., 2024). However, a known limitation of single-codebook methods is code underutilization, where many code vectors are rarely used, which can diminish the model's representational ability. In this paper, we address this issue by proposing a codebook pruning method applied to a residual vector quantization framework, enhancing representational ability while further reducing communication costs.

## 3 METHODOLOGY

We consider a V2X communication system illustrated in Fig. 1 where a central infrastructure extracts BEV feature maps from its LiDAR point cloud, while multiple vehicles generate corresponding feature maps from multi-view camera images. Then vehicles progressively encode their features into several index matrices using a residual quantization method with multiple codebooks, and transmit these indices to the infrastructure. Subsequently, the infrastructure decodes the shared feature maps and fuses the camera-based BEV feature maps with its own LiDAR-based ones via a multiscale fusion method. Finally, these fused feature maps are processed through an output head, and the infrastructure broadcasts the final perception results back to the connected vehicles.

### 3.1 V2I MULTIMODAL COOPERATIVE PERCEPTION FRAMEWORK

**Camera pipeline:** Vehicles are equipped with multiple cameras in various orientations that includes positions and optical parameters defined by extrinsic and intrinsic matrices. Each camera captures an image $\mathsf{I} \in \mathbb{R}^{H_{[i]} \times W_{[i]} \times 3}$, where $H_{[i]}$ and $W_{[i]}$ are the image height and width, and 3 represents the RGB channels. A ResNet-34 (He et al., 2016) backbone is used to extract image features at multiple scales, which are then input into the CVT module. Within the CVT module, learnable BEV map embeddings function as queries $Q$, while the multi-view features serve as the keys $K$ and values $V$, which are combined with a positional embedding that is calculated from the camera's extrinsic and intrinsic matrices. The final BEV feature map $\mathsf{F}_{[v]} \in \mathbb{R}^{H_{[v]} \times W_{[v]} \times C}$ with $C$ channel is obtained after several cross-view transform and downsampling operations.

**LiDAR pipeline:** The infrastructure is installed a LiDAR sensor to capture raw point clouds data, then preprocesses this data to a BEV map within a defined range. A PIXOR (Yang et al., 2018) backbone is used to process this BEV map, converting it into a 2D pseudo-image by flattening

the height dimension. This process projects the original data into a BEV feature map, denoted as $\mathbf{F}_{[r]} \in \mathbb{R}^{H_{[r]} \times W_{[r]} \times C}$. Here, $H_{[r]}$, $W_{[r]}$ and $C$ represent the height, width, and channels of the feature map, respectively.

## 3.2 CODEBOOK TRAINING AND PRUNING

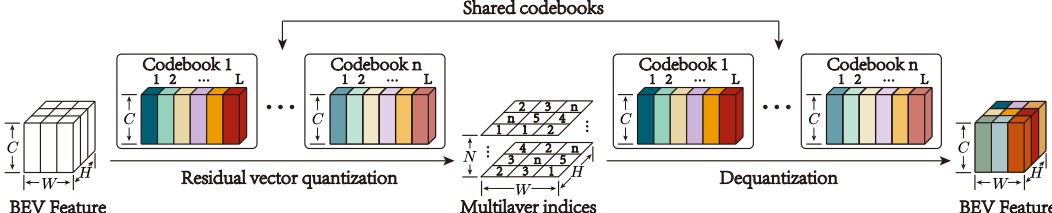

Figure 2: The process of residual codebook training.

To efficiently transmit the BEV feature map $\mathbf{F}_{[v]}$ from the vehicle with low communication costs, we employ a residual quantization method derived from the Vector Quantized Variational Autoencoder (VQ-VAE) (Van Den Oord et al., 2017). This approach uses multiple codebooks to progressively encode the feature map into a set of compact index matrices. The objective of this process is to approximate the continuous vectors in the feature map using a finite set of learnable vectors (i.e., codes) from shared codebooks. Then the infrastructure reconstructs the BEV feature map from transmitted index matrices based on shared codebooks.

### 3.2.1 CODEBOOK TRAINING

Codebook learning is an effective vector quantization method that enhances representational ability by training a codebook from a finite set of vectors. However, this approach is prone to code underutilization, where many codes are rarely or never selected during the quantization process (Wu & Yu, 2019). RVQ is a promising solution to break through the limitations of a single codebook, which aims to train multiple codebooks for approximating feature vectors recursively (Lee et al., 2022). As illustrated in Fig. 2, the residual quantization process involves vehicles and the infrastructure sharing a set of $N$ codebooks $\mathbb{D} = \{D_{[n]}\}_{n=1}^N$ for quantization and dequantization BEV feature maps, respectively. Each codebook $D_{[n]} \in \mathbb{R}^{C \times L}$ consists of $L$ code vectors that denoted as $D_{[n]l}, l \in [1, L]$. The objective for the first codebook $D_{[n]}$ is to minimize the Euclidean distance between the BEV feature vector $\mathbf{F}_{[v]h,w}$ at $h$-th and $w$-th position with the nearest code vector $D_{[n]l^*}$, written as,

$$Q(\mathbf{F}_{[v]h,w}; D_{[n]}) = D_{[n]l^*}, \text{ where } l^* = \underset{l \in 1,\dots,L}{\operatorname{argmin}} \|\mathbf{F}_{[v]h,w} - D_{[n]l}\|_2^2, \tag{1}$$

where $Q(\mathbf{F}_{[v]h,w}; D_{[n]})$ denotes the result of feature vector $\mathbf{F}_{[v]h,w}$ quantized by the codebook $D_{[n]}$. In residual quantization, this process is applied iteratively. The initial residual term is set to the input feature vector, $\boldsymbol{r}_{[1]} = \mathbf{F}_{[v]h,w}$. At each stage $n$, the current residual term $\boldsymbol{r}_{[n]}$ is quantized and the subsequent residual term $\boldsymbol{r}_{[n+1]}$ is computed as the remaining quantization error, expressed as,

$$\mathbf{r}_{[n+1]} = \mathbf{r}_{[n]} - Q(\boldsymbol{r}_{[n]}; D_{[n]}), \tag{2}$$

for $n = 1, 2, \cdots, N$. After $N$ stages, the original feature vector $\mathbf{F}_{[v]h,w}$ is represented by a set of $N$ indices $\{l_{[1]}^*, l_{[2]}^*, \cdots, l_{[N]}^*\}$. Consequently, the entire BEV feature map $\mathbf{F}_{[v]}$ is compressed into an integer tensor of indices $\mathbf{Z} \in \{1, \cdots, L\}^{H_{[v]} \times W_{[v]} \times N}$. This dramatically reduces the amount of data to be transmitted, as $N \ll C$ and integer indices can be encoded with fewer bits than floating-point vectors. Upon receiving the index tensor $\mathbf{Z}$, the infrastructure reconstructs the BEV feature vector $\hat{\mathbf{F}}_{[v]h,w}$ for each $h$ and $w$ position by summing the corresponding code vectors from each of the $N$ codebooks, as follows,

$$\hat{\mathbf{F}}_{[v]h,w} = \sum_{n=1}^N D_{[n]\mathbf{z}_{h,w,n}}, \ \forall \mathbf{Z}_{h,w,n} \in \{1, \cdots, L\}. \tag{3}$$

The training objective is to minimize the discrepancy between the original feature map $\mathbf{F}_{[v]h,w}$ and the quantized map $\hat{\mathbf{F}}_{[v]h,w}$ by optimizing the codebooks to minimize the quantization residual term at each stage. Thus, the commitment loss for codebook training could be formulated as,

$$\mathcal{L}_{\text{cmt}} = \sum_{h=1}^{H_{[v]}} \sum_{w=1}^{W_{[v]}} \|\mathbf{F}_{[v]h,w} - \text{sg}\Big[\sum_{n=1}^{N} D_{[n]}\mathbf{z}_{h,w,n}\Big]\|, \ \forall \ \mathbf{Z}_{h,w,n} \in \{1,\cdots,L\}, \tag{4}$$

where $\text{sg}[\cdot]$ denotes the stop-gradient operator, which prevents gradients from flowing back through the quantization process (Wu & Flierl, 2020). To sum up, the residual codebook representation offers two key advantages: i) It enables communication efficiency by transmitting low-bitrate integer indices instead of high-precision floating-point vectors; ii) It avoids code underutilization in the training for a single and large codebook by using multiple and smaller codebooks that are utilized more effectively.

### 3.2.2 CODEBOOK PRUNING

To further reduce the codebook size after training, which enables a more compact feature representation with fewer indices. We propose a codebook pruning method that involves two iterative steps: removing code vectors with low importance and combining highly similar code vectors. The importance score $s_l$ of the codebook $D_{[n]l}$ is defined as its usage frequency across all feature vectors, written as,

$$s_l = \sum_{h=1}^{H_{[v]}} \sum_{w=1}^{W_{[v]}} \mathbf{1}_{\text{condition}}\left(\mathbf{F}_{h,w} = D_{[n]l}\right), \tag{5}$$

where $\mathbf{1}_{\text{condition}}(\cdot)$ is the indicator function. A code vector $D_{[n]l}$ will be removed from the codebook $D_{[n]}$ if it with the minimum importance score among the codebook. Furthermore, the similarity $e_{a,b}$ between code vectors $D_{[n]a}, D_{[n]b} \in D_{[n]}$ could be evaluated by cosine similarity, i.e., $e_{a,b} = (D_{[n]a} \cdot D_{[n]b})/(\|D_{[n]a}\|\|D_{[n]b}\|)$. The pair with the maximum similarity score is replaced by a single new code, $D_{[n]c}$, computed as their average $D_{[n]c} = (D_{[n]a} + D_{[n]b})/2$. These removal and combination operations are performed iteratively until the number of code vectors in the codebook reaches a specified minimum, which serves as the stopping condition.

### 3.3 MULTISCALE FEATURE FUSION

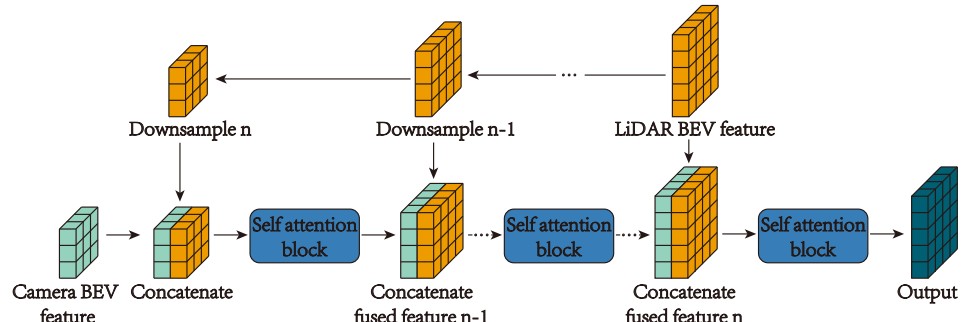

Figure 3: The process of multiscale feature fusion.

To conserve communication costs, vehicles significantly downsample their camera-based BEV feature maps $\mathbf{F}_{[v]} \in \mathbb{R}^{H_{[v]} \times W_{[v]} \times C}$ with a lower resolution, while it is challenge to directly fuse BEV feature maps with different spatial resolutions. To this end, we propose a multiscale fusion method that operates in a coarse-to-fine manner, as illustrated in Fig. 3. The LiDAR-based BEV feature map is progressively downsampled by a ratio of $\lambda$ at each stage, until its spatial resolutions match those of the camera-based BEV feature map $\mathbf{F}_{[v]}$ from the vehicle. The fusion process begins at concatenating these BEV feature maps and input to the self attention block for fusion, which consists of self attention, 3D convolution and upsamle modules. In self attention module, the concatenated feature are linearly projected into $\boldsymbol{Q}$, $\boldsymbol{K}$, and $\boldsymbol{V}$ tensors to produce a fused feature map $\mathbf{F}_{[f,m]}$ at stage $m$,

calculated as

$$\mathbf{F}_{[f,m]} = \text{softmax}(\frac{\boldsymbol{Q}\boldsymbol{K}^T}{\sqrt{\sigma}})\boldsymbol{V} \in \mathbb{R}^{\frac{H_{[r]}}{\lambda^m} \times \frac{W_{[r]}}{\lambda^m} \times C}, \tag{6}$$

where $\sigma$ is the key vector dimension used for scaling. A 3D convolutional module is applied to $\mathbf{F}_{[f,m]}$ to refine and align the aggregated feature maps for subsequent computation. The resulting feature maps is then upsampled by the ratio $\lambda$ and concatenated with the intermediate LiDAR-based feature map $\mathbf{F}_{[r]}$ from the corresponding scale. Then the concatenated feature map serves as the input for the self attention block in the next stage. This progressive fusion process is repeated $m$ stages, moving from the coarsest to the finest resolution to produce the full-resolution fused feature map.

### 3.4 LOSS FUNCTION

The final output of CoM-V2I is determined by a task-specific head, such as one for segmentation or detection.

**Segmentation task:** To handle class imbalance on map segmentation, we calculate the classification loss $\mathcal{L}_{\text{cls}}$ using a weighted Cross-Entropy metric to measure the difference between the predicted segmentation map $\mathbf{P} \in \mathbb{R}^{H \times W \times K}$ and the ground-truth map $\mathbf{Y} \in \mathbb{R}^{H \times W \times K}$ with a weight $\omega_k$ applied to each of the $K$ classes, expressed as,

$$\mathcal{L}_{\text{cls}} = -\sum_{h=1}^{H} \sum_{w=1}^{W} \sum_{k=1}^{K} \omega_k \mathbf{Y}_{h,w,k} \log(\mathbf{P})_{h,w,k}. \tag{7}$$

The total loss for the segmentation task $\mathcal{L}_{\text{seg}}$ is a weighted sum of the classification loss $\mathcal{L}_{\text{cls}}$ and the codebook commitment loss $\mathcal{L}_{\text{cmt}}$, that is,

$$\mathcal{L}_{\text{seg}} = \alpha_1 \mathcal{L}_{\text{cls}} + \alpha_2 \mathcal{L}_{\text{cmt}}, \tag{8}$$

where $\alpha_1$ and $\alpha_2$ are balancing weights.

**Detection task:** In contrast, the object detection task requires predicting various attributes for each bounding box, such as its dimensions (e.g., height, width, length) and heading angle. We use Smooth-L1 metric to calculate the regression loss $\mathcal{L}_{\text{reg}}$ for these attributes. By incorporating this regression component with weight $\alpha_3$, the total loss $\mathcal{L}_{\text{det}}$ for detection task could be written as

$$\mathcal{L}_{\text{det}} = \alpha_1 \mathcal{L}_{\text{cls}} + \alpha_2 \mathcal{L}_{\text{cmt}} + \alpha_3 \mathcal{L}_{\text{reg}}. \tag{9}$$

## 4 EXPERIMENTS

**Datasets and baselines**. We evaluate the effectiveness of the proposed CoM-V2I framework on the real-world V2X-Real dataset (Xiang et al., 2024) for object detection and the simulated V2X-Sim dataset (Li et al., 2022a) for map segmentation. To comprehensively assess our contributions, we compare our method against baselines organized into three groups: i) V2I cooperation framework. To evaluate the overall multi-modal strategy, we compare against foundational approaches, including Camera-only, LiDAR-only, and LiDAR2cam. ii) Communication efficiency. We compare proposed codebook-based communication method with Where2comm, How2comm, Codefilling, and Fullcomm baselines. iii) Multiscale fusion. To validate our fusion mechanism, we compare it against other state-of-the-art methods, such as V2I-Coop, HM-ViT, CoarseFusion, and BEVFusion. The details of these baselines are listed on Appendix A.1.2.

**Experimental settings.** During training for each frame, we designate the infrastructure as the ego-agent and only consider vehicles that are within a $60m$ communication range. We use ResNet-34 and PIXOR as the backbones for processing camera images and LiDAR point clouds, respectively. The camera-based BEV feature map is constructed from multi-view images by the FAX module (Xu et al., 2022a), resulting in a resolution of $32 \times 32 \times 128$. In contrast, the LiDAR-based BEV feature map has a higher resolution of $128 \times 128 \times 128$. The residual vector quantization process employs three codebooks, each containing 128 code vectors. All models were trained using the Adam optimizer with a cosine annealing learning rate scheduler for 100 epochs and 2 batchsize. Prior to evaluating the model's performance, we pre-compute and cache all camera-based BEV feature maps

from the entire dataset to accelerate the inference of codebook pruning. For the detection task, we use Average Precision (AP) calculated at Intersection over Union (IoU) thresholds of 0.3, 0.5, and 0.7. For the segmentation task, we evaluate performance using the mean Intersection over Union (mIoU) between the predicted map and the ground-truth map.

## 4.1 QUALITATIVE EVALUATIONS

Table 1: Performance comparison of CoM-V2I with other methods on two tasks.

| Method | Modality[†] | Detection (AP)[‡] | | | Segmentation (mIoU)[#] | | |
|---|---|---|---|---|---|---|---|
| | | AP@0.3 | AP@0.5 | AP@0.7 | Vehicle | Road | Lane |
| Camera-only(Xu et al., 2022a) | IC + VC | 42.71 | 29.52 | 15.34 | 10.97 | 51.87 | 21.19 |
| LiDAR-only(Wang et al., 2020) | IL + VL | **71.79** | 63.71 | 38.05 | 21.18 | 88.27 | 76.79 |
| LiDAR2cam | IC + VL | 63.07 | 60.18 | 47.62 | 15.78 | 76.61 | 51.32 |
| Where2comm(Hu et al., 2022) | IL + VC | 70.92 | 68.65 | 56.66 | 62.49 | 87.42 | 74.27 |
| How2comm(Yang et al., 2023a) | IL + VC | 70.14 | 68.15 | 55.18 | 61.42 | 91.87 | 74.46 |
| Codefilling(Hu et al., 2024) | IL + VC | 62.87 | 62.12 | 55.35 | 54.75 | 90.97 | 76.64 |
| Fullcomm | IL + VC | 70.49 | 68.96 | 58.11 | 63.52 | 92.10 | 77.37 |
| HM-ViT(Xiang et al., 2023) | IL + VC | 63.11 | 59.52 | 46.21 | 59.07 | **92.48** | 70.29 |
| V2I-Coop(Zhou et al., 2025) | IL + VC | 69.89 | 65.46 | 52.75 | 55.52 | 91.78 | 72.06 |
| BEVFusion(Liu et al., 2023b) | IL + VC | 67.43 | 63.42 | 46.28 | 18.26 | 87.42 | 66.04 |
| CoarseFusion | IL + VC | 69.48 | 66.37 | 52.15 | 39.18 | 90.75 | 77.07 |
| **CoM-V2I** | IL + VC | 71.25 | **69.14** | **58.66** | **63.56** | 91.70 | **77.56** |

[†] IC: Infrastructure Camera; IL: Infrastructure LiDAR; VC: Vehicle Camera; VL: Vehicle LiDAR.
[‡] "AP@0.3", "AP@0.5", and "AP@0.7" represent the AP at IoU thresholds of 0.3, 0.5, and 0.7.
[#] The mIoU is adopted for evaluating the segmentation task.

To fairly compare different baselines, we divided them into three groups described above. For framework, only input sources are different among compared baselines. For communication efficiency, compression methods for BEV feature maps are various. For fusion startegy, only fusion method are different. The FAX (Xu et al., 2022a) module and the PIXOR module are adopted in all baselines to represent the BEV feature maps of the cameras and LiDAR. As shown in Tab. 1, all LiDAR involved methods outperform the camera-only method, which highlights the limitation of purely camera-based CP. The LiDAR-only method achieves the highest AP at an IoU threshold of 0.3, though its performance drops at the stricter thresholds of 0.5 and 0.7. In the communication efficiency comparison, our CoM-V2I method achieves the best performance, matching or even surpassing the Fullcomm baseline (which transmits uncompressed features). This result indicates that our residual vector quantization-based approach effectively enhances feature representation. Regarding the fusion strategy, our multiscale method outperforms both fine-grained and coarse-grained fusion baselines, achieving a competitive trade-off between performance and communication cost. In summary, CoM-V2I obtains the highest AP at the 0.5 and 0.7 IoU thresholds for object detection and the best mIoU for the vehicle and lane segmentation classes.

## 4.2 COMMUNICATION COSTS

Fig. 4 compares our proposed CoM-V2I with other communication-efficient methods, evaluating the trade-off between communication costs and performance. The performance of all algorithms drops significantly without the camera-based BEV feature maps, which highlights the importance of CP. Our CoM-V2I method exhibits two key advantages. i) achieves the best performance across all three IoU thresholds at similar communication costs, with an AP improvement of $1\%$ to $6\%$ over other methods. ii) significantly reduces communication costs while maintaining competitive performance, especially outperforming the baseline that without an efficient communication strategy. This strong performance can be attributed to the effective feature representation from our residual vector quantization, while the proposed codebook pruning further decreases communication costs by removing less-utilized and combining high-similarity code vectors. A detailed analysis of the mean absolute error (mAE) between the original and quantized BEV feature maps, along with their corresponding communication costs and performance, is provided in Tab. 5 in Appendix A.2.

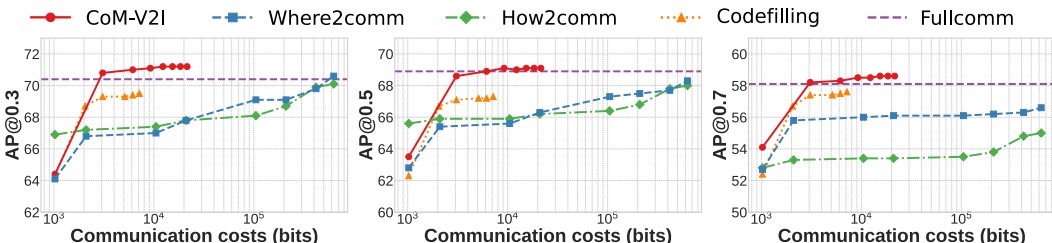

Figure 4: AP and mIoU for the vehicle class versus communication costs. The communication costs for CoM-V2I and Codefilling is calculated by $H_{[v]} \times W_{[v]} \times \sum_{n=1}^{N} \lceil \log_2(|D_{[n]}|) \rceil$, where $|\cdot|$ denotes the size of $D_{[n]}$. For Where2comm and How2comm, costs is determined by multiplying their respective compression ratios by the size of the original BEV feature map. To make the scale smooth, we use a cost of 1024 to represent no feature maps are transmitted.

## 4.3 QUALITATIVE RESULTS

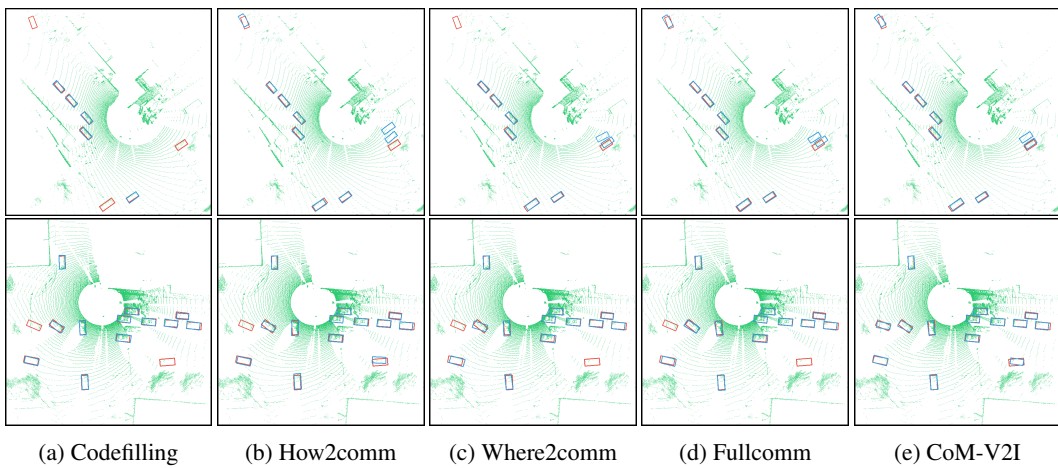

| (a) Codefilling | (b) How2comm | (c) Where2comm | (d) Fullcomm | (e) CoM-V2I |

Figure 5: Qualitative comparison of two scenarios from the V2X-Real dataset, where a different infrastructure is designated as the ego agent in each case. Red bounding boxes represent the ground truth, while blue bounding boxes represent the predictions.

Fig. 5 presents a qualitative comparison of several communication-efficient baselines. The results highlight two common failure modes in existing methods. Some methods, such as Codefilling, produce accurate bounding boxes for detected objects but fail to identify all objects. In contrast, other methods, like How2comm, detect almost all objects, but their predicted bounding boxes suffer from localization inaccuracies. Our CoM-V2I method successfully bounds all objects while also predicting their relevant attributes with high precision. Additional qualitative results, including map segmentation, are shown in Appendix A.3.

## 4.4 ABLATION RESULTS

Table 2: Performance comparison with and without codebook pruning. The model without pruning were trained from scratch with the defined codebook size, while with pruning started with three large, pre-trained codebooks of 128 code vectors that was subsequently pruned to the target size.

| Codebook pruning | Codebook size | Detection | | | Segmentation | | |
|---|---|---|---|---|---|---|---|
| | | AP@0.3 | AP@0.5 | AP@0.7 | Vehicle | Road | Lane |
| ✗ | 2, 2, 2 | 67.34 | 66.16 | 56.62 | 60.98 | 90.92 | 68.95 |
| ✓ | 2, 2, 2 | 70.82 ▲3.48 | 68.58 ▲2.42 | 58.16 ▲1.54 | 63.52 ▲2.54 | 91.74 ▲0.82 | 75.69 ▲6.74 |
| ✗ | 4, 4, 4 | 69.91 | 67.68 | 57.50 | 61.87 | 91.56 | 73.32 |
| ✓ | 4, 4, 4 | 71.03 ▲1.12 | 68.87 ▲1.19 | 58.34 ▲0.84 | 63.52 ▲1.65 | 91.74 ▲0.18 | 75.69 ▲2.37 |

**Pruning efficiency.** Tab. 2 presents a performance comparison between codebooks created via pruning versus those trained from scratch. Based on our finding in Fig. 4 that pruned codebooks achieve competitive performance at small sizes, we focus this analysis on codebooks with just 4 and 2 code vectors. Pruning codebooks are derived by pruning a large, pre-trained codebook (originally with 128 code vectors), while unpruned versions are trained from scratch with the smaller target size. The results demonstrate that the pruning method outperforms the training from scratch method with achieving AP improvement of approximately 0.01 to 0.02 at each IoU threshold.

**Vehicles disconnection.** In the inference process, the transmitted BEV feature maps from vehicles are zeroed out in proportional samples according to the 'vehicles disconnected ratio'. Tab. 3 shows the AP results at various vehicle disconnection ratios. The performance degrades slightly with the disconnected ratio increases. For instance, as the disconnection ratio increases from 0.1 to 0.9, the AP drops by only 5% at the stricter 0.5 and 0.7 IoU thresh-

Table 3: Robustness to vehicle disconnection.

| Ratio | AP@0.3 | AP@0.5 | AP@0.7 |
|---|---|---|---|
| 0.1 | 70.63 ▼ 0.57 | 68.68 ▼ 0.42 | 58.45 ▼ 0.15 |
| 0.3 | 69.07 ▼ 2.13 | 67.51 ▼ 1.59 | 57.52 ▼ 1.08 |
| 0.5 | 67.74 ▼ 3.46 | 66.39 ▼ 2.71 | 56.26 ▼ 2.34 |
| 0.7 | 66.11 ▼ 5.09 | 65.03 ▼ 4.07 | 55.20 ▼ 3.40 |
| 0.9 | 65.08 ▼ 6.12 | 64.05 ▼ 5.05 | 54.67 ▼ 3.93 |

olds. This trend highlights the robustness of CoM-V2I to communication failures from individual vehicles.

**BEV feature channels and resolutions.** We evaluate the performance of CoM-V2I with different feature channels and resolutions, as shown in Tab. 4. The results indicate that AP at each IoU threshold decreases as the number of channel is reduced, which demonstrates that the representational capacity of residual quantization method diminishes with fewer channels. While the performance drop is slight at the 0.3 IoU threshold, it becomes more significant (approximately 0.1) at the stricter 0.7 threshold. Regarding feature resolution, the model maintains competitive performance even as the resolution is reduced, with AP dropping by only 0.01 to 0.02 at the 0.3 and 0.5 IoU thresholds. This trend demonstrates that our multiscale fusion method can effectively handle BEV feature maps of various resolutions.

Table 4: Ablation study on BEV feature channels and resolutions. For the channel evaluation, the camera-based and LiDAR-based BEV feature map resolutions are fixed at $32 \times 32$ and $128 \times 128$, respectively. For the resolution evaluation, the number of channels is fixed at 128, while the resolution of the camera-based BEV feature maps is varied.

| Channels | AP@0.3 | AP@0.5 | AP@0.7 | Resolutions | AP@0.3 | AP@0.5 | AP@0.7 |
|---|---|---|---|---|---|---|---|
| 32 | 67.31 | 63.87 | 48.20 | $8 \times 8$ | 69.41 | 67.29 | 58.15 |
| 64 | 68.43 | 64.81 | 52.64 | $16 \times 16$ | 69.79 | 67.91 | 58.25 |
| 96 | 69.91 | 67.67 | 56.13 | $32 \times 32$ | **71.25** | **69.14** | **58.66** |
| 128 | **71.25** | **69.14** | **58.66** | $64 \times 64$ | 70.08 | 67.23 | 57.54 |

## 5 CONCLUSION AND DISCUSSION

In this paper, we introduced CoM-V2I, an economical and communication-efficient V2I cooperative perception framework. To enhance economic viability, we proposed a multimodal cooperation paradigm where the infrastructure is equipped with a high-cost LiDAR, while vehicles use more affordable cameras. We introduced a codebook pruning method for residual vector quantization to significantly decrease communication costs. Furthermore, to handle the low-resolution features transmitted by vehicles for bandwidth conservation, we presented a multiscale fusion method capable of fusing multi-modal BEV feature maps at various resolutions. Experiments on both the real-world V2X-Real and simulated V2X-Sim datasets demonstrate that CoM-V2I outperforms previous state-of-the-art methods on both detection and segmentation tasks.

**Limitation and future work.** The results from Tab. 3 reveal that the performance of CoM-V2I not drops dramatically when BEV feature maps from vehicles transmitted fail. However, a key limitation is its reliance on the central infrastructure due to performance drops significantly if the infrastructure is disconnected during inference, as illustrated in Appendix A.2. In the future, we plan to address this by recovering lost feature maps from the infrastructure for enhancing the robustness.

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

## A  APPENDIX

LLM USAGE DECLARATION

We acknowledge the use of a Large Language Model (LLM) for assistance with grammar, phrasing, and LaTeX formatting.

## A.1 IMPLEMENTATION DETAILS

### A.1.1 DATASETS

**V2X-Real dataset** (Xiang et al., 2024) comprises 5772 training, 1253 test, and 717 validation frames. These frames were collected from 63 diverse scenarios involving a total of two infrastructures and two vehicles, with each frame containing data from both infrastructures and at least one vehicle. Every agent is equipped with a LiDAR sensor, providing a point cloud, and either 2 or 4 cameras that collectively offer a $360°$ horizontal field of view.

**V2X-Sim dataset** (Li et al., 2022a) is based on the nuScenes format and contains 100 driving scenes with 100 frames each (Caesar et al., 2020). Every frame includes one infrastructure and up to five vehicles. Each agent is equipped with a LiDAR sensor and multiple cameras deployed surrounding the agent. To facilitate training with the OpenCOOD toolkit (Xu et al., 2022b), which natively supports the V2X-Real format, we reorganized the V2X-Sim dataset's structure to match that of V2X-Real. In addition, we divided 80 scenes for training and 20 scenes for validation and inference.

### A.1.2 BASELINES

We organize compared baselines into three groups to evaluate the different aspects of our framework.

- **Modality in framework:** i) Camera-only. Vechiles represent BEV feature maps based on images from 4 multi-view cameras, while the infrastructure is equipped with 2 multi-view cameras (Xu et al., 2022a). ii) LiDAR-only. All agents use point cloud data from the LiDAR to CP with a consistent range of $[-51.2, -51.2, -3, 51.2, 51.2, 3]$ meters (Wang et al., 2020). iii) LiDAR2cam. The infrastructure only adopts multi-view cameras for CP, while vehicles use their LiDAR sensors. For a fair comparison, our proposed communication and fusion methods are applied to all modality baselines.

- **Communication efficiency:** i) Where2comm. Camera-based BEV feature maps are compressed into spatial confidence maps by filtering out non-critical feature vectors (Hu et al., 2022). ii) How2comm. A mutual information-aware communication mechanism and a spatial-channel filtering method are used to compress BEV feature maps (Yang et al., 2023a). iii) Fullcomm. This baseline does not employ any communication efficiency mechanism. iv) Codefilling. A single codebook is trained to represent BEV feature maps and transmits a compact index matrix (Hu et al., 2024). The fusion method and modality for these baselines are aligned with the CoM-V2I.

- **Fusion strategy:** i) HM-ViT. The camera-based BEV feature map is upsampled to a $128 \times 128 \times 128$ resolution to align with the LiDAR-based feature map for same scale fusion (Xiang et al., 2023). ii) V2I-Coop. A cross-attention mechanism is used for feature fusion, followed by local and global self-attention modules (Zhou et al., 2025). iii) CoarseFusion. The LiDAR-based BEV feature map is downsampled to a $32 \times 32 \times 128$ resolution to facilitate coarse-grained feature fusion between modalities. iv) BEVFusion. BEV feature maps are first resized to a uniform height and width, then concatenated along the channel dimension and passed through convolutional layers for fusion (Liu et al., 2023b). The proposed communication method are performed for each baseline.

### A.1.3 TRANING STRATEGY

Following previous works (Xu et al., 2022a; Xiang et al., 2023), all models are trained using the AdamW optimizer with a cosine annealing scheduler and a learning rate of $2 \times 10^{-4}$. For the object detection task, we train models for 100 epochs with a batch size of 2. Within the classification loss, we apply weights of 1.0 for negative samples and 25.0 for positive samples. The total loss components $\mathcal{L}_{cls}$, $\mathcal{L}_{cmt}$, and $\mathcal{L}_{reg}$ are balanced with weights $\alpha_1$, $\alpha_2$, and $\alpha_3$ set to 5.0, 2.0 and 1.0, respectively. For the map segmentation task, models are trained for 80 epochs with a batch size of 1. We employ a focal loss (Lin et al., 2017) for the classification loss $\mathcal{L}_{cls}$ to address class imbalance. The balancing weights $\alpha_1$ and $\alpha_2$ set to 2.0 and 5.0, respectively.

## A.2 ADDITIONAL ABLATION STUDIES

**Reconstruction errors for BEV feature maps.** Tab. 5 represents the AP and mIoU is only slightly reduced even with a higher mAE in feature reconstruction, which demonstrates that many vectors in the original features are redundant. The codebook pruning method effectively removes less utilized codes vectors without a significant loss in performance.

Table 5: Performance of CoM-V2I in various mAE and communication costs. The mAE is used for evaluating the difference between the original and quantized camera-based BEV feature map.

| Costs (bits) | Object detection | | | | Map segmentation | | | |
|---|---|---|---|---|---|---|---|---|
| | mAE | AP@0.3 | AP@0.5 | AP@0.7 | mAE | Vehicle | Road | Lane |
| 21504 | $1.20 \times 10^{-3}$ | 71.25 | 69.14 | 58.66 | $2.30 \times 10^{-6}$ | 63.56 | 91.77 | 75.56 |
| 15360 | $1.37 \times 10^{-3}$ | 71.23 | 69.14 | 58.63 | $3.80 \times 10^{-6}$ | 63.54 | 91.75 | 75.69 |
| 9216 | $2.20 \times 10^{-3}$ | 71.16 | 69.06 | 58.55 | $2.06 \times 10^{-5}$ | 63.52 | 91.74 | 75.69 |
| 6144 | $3.31 \times 10^{-3}$ | 71.03 | 68.87 | 58.34 | $2.42 \times 10^{-5}$ | 63.52 | 91.74 | 75.69 |
| 3072 | $6.93 \times 10^{-3}$ | 70.82 | 68.58 | 58.16 | $9.50 \times 10^{-5}$ | 63.52 | 91.74 | 75.69 |
| 0 | - | 64.49 | 63.51 | 54.10 | - | 59.51 | 90.36 | 74.41 |

**Component analysis.** The results in Tab. 6 demonstrate that both our codebook pruning and multiscale fusion methods improve model performance. Specifically, compared to a baseline with a single codebook, incorporating RVQ improves the AP by $8\%$, $4\%$, and $2\%$ at IoU thresholds of $0.3$, $0.5$, and $0.7$, respectively. Furthermore, applying our codebook pruning strategy maintains performance comparable to that of the original full codebook with lower communication costs. In the end, the multiscale fusion method achieves $14\%$ improvement in AP at IoU $0.7$.

Table 6: Component ablation. "Prune" and "MSF" refer to codebook pruning and multiscale fusion.

| MSF | Prune | AP@0.3 | AP@0.5 | AP@0.7 |
|---|---|---|---|---|
| | | 70.45 | 63.42 | 41.79 |
| ✓ | | 71.25 | 69.14 | 58.66 |
| | ✓ | 69.78 | 62.94 | 41.40 |
| ✓ | ✓ | 70.82 | 68.58 | 58.16 |

**Infrastructure disconnection.** In contrast to the robustness of vehicle disconnection, COM-V2I is more sensitive to losing the infrastructure's BEV feature maps, which are zeroed out based on the 'infrastructure disconnection ratio'. As shown in Tab. 7, the AP drops by approximately $12\%$ at each IoU threshold for every $0.2$ increase in the infrastructure disconnection ratio. Despite this, even when the disconnection ratio still surpasses that of the camera-only method, as shown on Tab. 1. This demonstrates that LiDAR and camera fusion method offers an effective trade-off between system cost and reliability.

Table 7: AP under various infrastructure disconnection ratios.

| Ratio | AP@0.3 | AP@0.5 | AP@0.7 |
|---|---|---|---|
| 0.1 | 64.68▼6.52 | 62.55▼6.55 | 54.33▼4.27 |
| 0.3 | 52.01▼19.19 | 50.32▼18.78 | 44.59▼14.01 |
| 0.5 | 38.45▼32.75 | 36.87▼32.32 | 33.31▼25.29 |
| 0.7 | 25.52▼45.68 | 24.04▼45.06 | 22.26▼36.34 |
| 0.9 | 13.09▼58.11 | 11.81▼57.29 | 9.74▼48.86 |

## A.3 VISUALIZATION

**Object detection.** Fig. 6 shows the visual comparison of object detection between CoM-V2I and other baselines on V2X-Real dataset. CoM-V2I outperforms others on precisely bounding all objects and predicting relevant attributes.

**Map segmentation.** Qualitative results for the map segmentation task on the V2X-Sim dataset are shown in Fig. 7. The visualizations demonstrate that CoM-V2I accurately segments the vehicle class across various driving scenes.

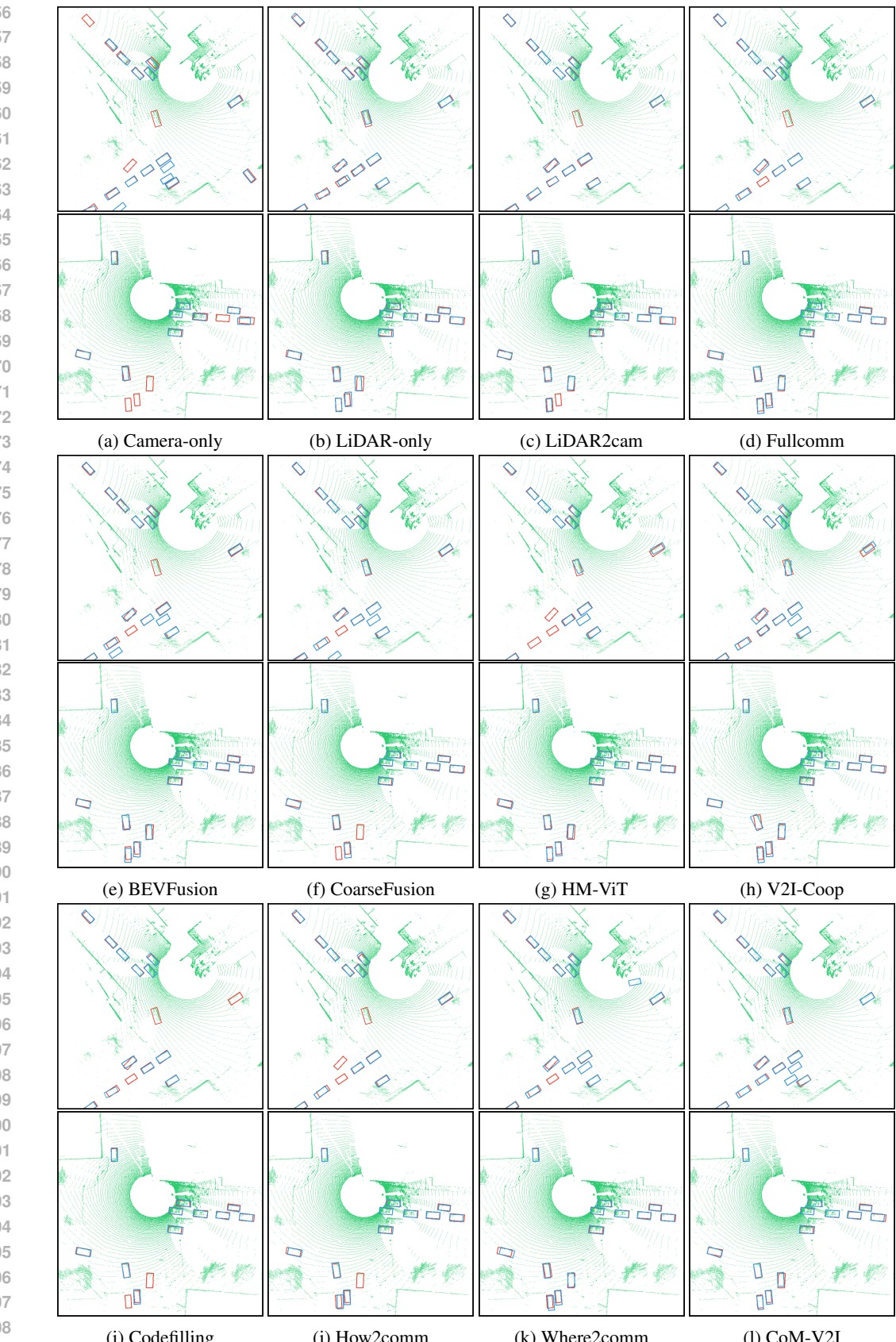

Figure 6: Visual results of all baselines in object detection. The Red and blue bounding boxes represent the ground truth and prediction, respectively.

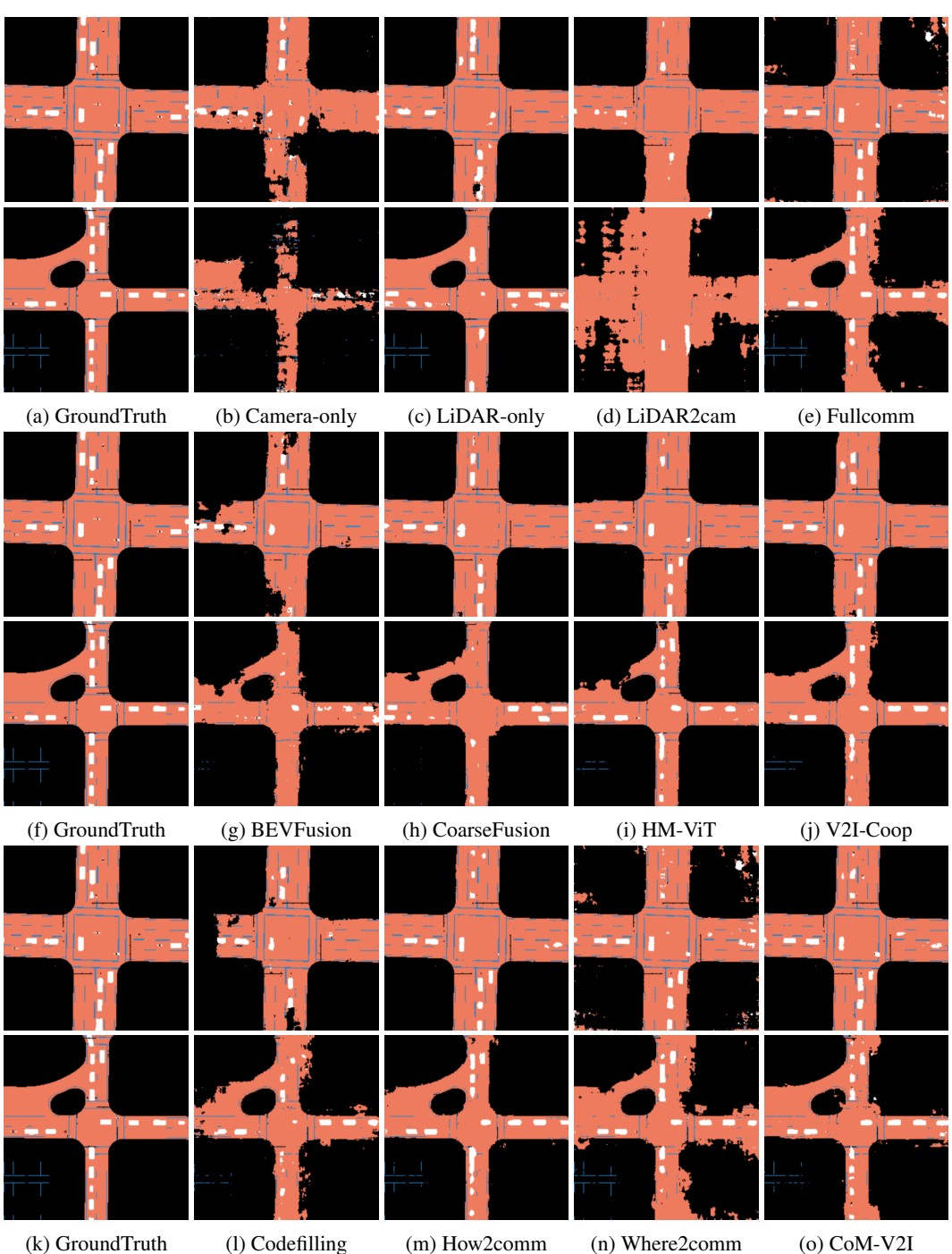

Figure 7: Qualitative results for map segmentation. The white, orange and blue color map represent vehicle, road and lane classes, respectively.

