# OpenReview forum: "CoM-V2I: Communication-Efficient Multimodal Cooperative Perception via Codebook Pruning and Multiscale Fusion"
_ICLR.cc/2026/Conference — ICLR 2026 Conference Withdrawn Submission_

### Official Review · Reviewer_tJVz · 2025-10-28

**Soundness:** 2
**Presentation:** 2
**Contribution:** 2
**Rating:** 2
**Confidence:** 4

**Summary:**

This paper proposes CoM-V2I, a novel framework for Communication-efficient Multimodal Vehicle-to-Infrastructure (V2I) cooperative perception. It enhances communication efficiency through a residual vector quantization-based codebook representation method and further reduces costs with a codebook pruning technique. Experimental results demonstrate that CoM-V2I surpasses existing baselines in both perception performance and communication efficiency. However, the proposed method primarily constitutes engineering improvements on existing techniques, leading to a slight lack of novelty.

**Strengths:**

1.Improved Codebook Utilization: The proposed residual vector quantization-based codebook representation method leverages multiple smaller codebooks, effectively mitigating the issue of under-utilization commonly observed in the training of a single, large codebook.

2.Reduced Communication Bandwidth: The codebook pruning technique further reduces communication bandwidth requirements by achieving a more compact feature representation using fewer indices.

3.Good Performance-Communication Trade-off: CoM-V2I achieves good performance in balancing perception accuracy with communication bandwidth efficiency, which is crucial for practical deployment.

**Weaknesses:**

1.Limited Novelty in Core Components: The residual vector quantization-based codebook representation and the codebook pruning methods presented in this paper appear to be engineering improvements on existing techniques, particularly drawing parallels with Codefilling, thus exhibiting a slight lack of innovation. Furthermore, the multiscale feature fusion module, which combines FPN with self-attention, is a standard design and does not seem to incorporate specific considerations for cooperative perception scenarios.

2.Focus on Infrastructure-Centric Perception vs. Vehicle-Centric Needs: This work focuses on sharing features from connected vehicles to the infrastructure for feature fusion and environmental perception, with the infrastructure then broadcasting the perception results within a fixed range back to the vehicles. However, in practical autonomous driving scenarios, the environmental perception results around the ego vehicle are often of paramount concern, a point seemingly overlooked by this work. For instance, if vehicle agent i and agent j have significant overlapping perception ranges, but both have minimal overlap with infrastructure agent k, the proposed operational flow seems to have limited utility in enhancing agents i and agent j’s perception of their immediate surroundings. Therefore, it is recommended that the authors evaluate the network with vehicles as ego-agents to demonstrate the effectiveness of cooperative perception in practical deployment scenarios.

**Questions:**

See the weaknesses.

---

### Official Review · Reviewer_gAPv · 2025-10-30

**Soundness:** 3
**Presentation:** 2
**Contribution:** 2
**Rating:** 4
**Confidence:** 3

**Summary:**

This paper proposes a novel communication-efficient collaborative perception framework. The main contributions include: (i) a vector quantization-based compression module with codebook pruning; and (ii) a multi-scale feature fusion module. Experimental results demonstrate that the proposed approach achieves a favorable trade-off between perception performance and communication bandwidth on both real-world and simulated datasets.

**Strengths:**

1. The paper is well organized and clearly written.
2. The evaluation is comprehensive, covering both real-world and simulated datasets as well as detection and segmentation tasks.

**Weaknesses:**

1. Sensor generalization. The paper assumes that infrastructure nodes are equipped with high-resolution LiDAR sensors, while vehicles use cost-effective multi-view cameras. It remains unclear how the proposed method generalizes to other sensing modalities. Is the framework limited to this specific sensor configuration?
2. Effectiveness of codebook pruning. Codebook pruning introduces a potentially lossy operation that modifies the code space. This may create a domain gap between training and inference.
3. Weak connection between the two innovations. The proposed multi-scale fusion module appears to be weakly related to the codebook pruning mechanism and communication efficiency. The overall narrative might be strengthened by clarifying how these two components interact or complement each other.
4. Lack of ablation on multi-scale fusion. The effectiveness of the multi-scale fusion design is not sufficiently validated through ablation studies.

**Questions:**

1. Can you quantify any domain shift introduced by codebook pruning between training and inference—reporting accuracy vs. pruning ratio?
2. What is the concrete interaction between the multi-scale fusion module and codebook pruning in delivering accuracy–communication gains? Please provide ablations isolating each component？

---

### Official Review · Reviewer_9G1Z · 2025-10-31

**Soundness:** 2
**Presentation:** 2
**Contribution:** 2
**Rating:** 4
**Confidence:** 4

**Summary:**

The paper proposes CoM-V2I, a communication-efficient heterogeneous cooperative perception framework. It introduces a residual vector quantization (RVQ)-based codebook representation method that compresses BEV feature maps into lightweight indices before transmission. A codebook pruning strategy is further proposed to reduce the codebook size by removing low-importance and merging high-similarity code vectors, thereby decreasing communication costs with minimal impact on perception performance. In addition, a multiscale fusion mechanism is designed to progressively integrate multimodal BEV features from both infrastructure and vehicles in a coarse-to-fine manner.

**Strengths:**

The paper has a clear motivation: it aims to address the trade-off between perception accuracy and communication bandwidth in heterogeneous multi-agent cooperative perception systems.

The idea of combining multiple codebooks with pruning is well-motivated and effectively improves communication efficiency while maintaining strong performance.

Experiments are relatively comprehensive and demonstrate that the proposed framework achieves a good accuracy–bandwidth trade-off in heterogeneous V2I settings.

**Weaknesses:**

The novelty is somewhat limited. The overall structure appears to combine elements of BEVFusion and CodeFilling, where BEV features are quantized via VQ-VAE-like multi-codebook representations.

The proposed codebook pruning strategy is relatively simple, and the multiscale fusion design resembles commonly used coarse-to-fine fusion approaches in prior work.

The experimental setup is simplified — each agent uses only one sensor modality, and it remains unclear how the proposed framework adapts to truly heterogeneous modalities or scales to multi-agent scenarios beyond the V2I case.

**Questions:**

see Weaknesses

---

### Note · Authors · 2025-11-21

I have read and agree with the venue's withdrawal policy on behalf of myself and my co-authors.